# Estimation for Refined Carbon Storage of Urban Green Space and Minimum Spatial Mapping Scale in a Plain City of China

**Nan Li** [1,2]**, Liang Deng** [1,2]**, Ge Yan** [1,2,3]**, Mengmeng Cao** [1,2] **and Yaoping Cui** [1,2,*]

[1]   Key Laboratory of Geospatial Technology for the Middle and Lower Yellow River Regions, Henan University, Ministry of Education, Kaifeng 475004, China; linan0716@henu.edu.cn (N.L.)
[2]   College of Geography and Environmental Science, Henan University, Kaifeng 475004, China
[3]   Institute of Geographic Science and Natural Resources Research, Chinese Academy of Sciences, Beijing 100101, China
**\***   Correspondence: cuiyp@lreis.ac.cn

**Abstract:** Current cities are not concrete jungles and deserts with sparse vegetation. Urban green space (UGS) appears widely in human activity areas and plays an important role in improving the human living environment and accumulates carbon storage. However, given the scattered distribution of UGS, studies on both the refined spatial estimation of carbon storage and appropriate mapping scale are still lacking. Taking the downtown area of Kaifeng, China, as the study area, this study verified the i-Tree Eco model on the basis of a field survey and accurately estimated the spatial carbon storage of UGS by combining it with remote sensing data, and finally, we obtained the minimum spatial mapping scale of UGS carbon storage by scaling. The results showed that (1) the total area of UGS in study area was 26.41 km$^2$, of which the proportion of total area of residential area and park green spaces was about 50%. The area of UGS per capita in the study area is 40.49 m$^2$. (2) Within the 123 survey samples, the proportion of communities with tree–shrub–herbs structure was the highest, 51.22%. The average carbon density was 5.89 kg m$^{-2}$, among which the park, protective and square green spaces had the highest carbon density in all land use types. (3) The total carbon storage of UGS in the study area was 114,389.17 t, and the carbon storage of UGS per capita was 175.39 kg. Furthermore, the scaling analysis showed that 0.25 km spatial resolution was the minimum spatial scale for UGS carbon storage mapping. This study improves our understanding of urban carbon storage, highlights the role and potential of UGS in carbon neutrality, and clarifies the importance of estimating urban carbon storage at appropriate scales. This study is also of great significance for rationally understanding the terrestrial carbon cycle in urban areas and improving regional climate simulations.

**Keywords:** urban green space; green space classification; carbon storage; spatial mapping scale; i-Tree model

## 1. Introduction

Urban green space (UGS) is an important part of urban ecosystems and has become an important content for building a better urban living environment worldwide [1,2]. An UGS is public open space that serves the public, and many UGSs have developed into ecological systems for connecting entire cities, which is also an important factor to be considered during urban planning [3]. Refined study of UGS carbon storage is of great significance for correctly understanding the regulatory effects of human gathering cities on the regional climate [4]. Although many studies estimate UGS carbon storage from vegetation typology and using remote sensing, there is still a lack of relevant research on UGS carbon storage accounting at appropriate spatial mapping scales [5].

The study on accounting carbon storage of UGSs has been paid more and more attention [6,7]. Some studies have evaluated the above-ground carbon storage of UGS in Leipzig, Europe, and pointed out that urbanization had an important impact on the

UGS carbon storage [8]. The application of remote sensing data also provides strong support for the estimation of large-scale UGS carbon storage. The correlation between UGS carbon storage and the remote sensing vegetation index has been confirmed, and regression models have been established to account for the UGS carbon storage [9,10]. In terms of model simulations, the U.S. Forest Service developed the i-Tree model and City-green model for studying the carbon sequestration benefits of urban vegetation, which have been globally used [11–14]. The i-Tree model requires many sample surveys and a large workload [15,16]. The City-green model uses stormwater runoff, carbon sequestration and air pollution removal potential as the ecological parameters, but it ignores tree species composition, which has a certain impact on the result accuracy [17,18]. In fact, different carbon storage estimation methods have their own advantages and disadvantages (Table 1). Therefore, when various methods are used, the localization of model and scale issue of regional expansion application deserve special attention.

Refined estimation of the UGS carbon storage in specific cities is necessary, since field survey data may have greater impacts on the accuracy of carbon storage than models. At the same time, the appropriate spatial scale must be considered in the process of urban carbon storage spatial mapping. The UGS carbon density is affected by green space spatial layout, urban development intensity, green space community structure and other factors, and all of these result in differences in the UGS carbon density among different cities [19,20]. Additionally, due to factors such as the construction policies and human disturbances, UGS in different regions has great differences in spatial morphology and species composition. All these factors increase the difficulty of studying UGS carbon storage [21]. Therefore, the combination of field survey and remote sensing methods is particularly important. Studies on urban vegetation carbon storage have been reported with the field survey data and remote sensing images in several cities like Shenyang, China and Seattle, US [22,23]. These results highlight the importance of urban vegetation carbon storage for reducing the local atmospheric $CO_2$ concentration and mitigating climate change pressures. Overall, limited by factors such as the wide distribution range and spatial heterogeneity of UGS, field survey data are indispensable for accurately estimating the UGS carbon storage, and combining remote sensing can effectively expand the study scale [24]. When combining field survey data with remote sensing images for studying, an important issue arises, namely, the appropriate scale of spatial mapping for UGS carbon storage [25]. However, there is currently a lack of exploration on the appropriate scale of UGS carbon storage, and the appropriate mapping scale will directly restrict the accuracy of the UGS carbon storage estimation [5,26,27]. Therefore, an appropriate or minimum scale is particularly important for the spatial estimation and mapping of urban carbon storage.

**Table 1.** Methods for urban carbon storage estimation.

| Methods | Theories | Required Data | Advantage | Disadvantage | Reference |
|---|---|---|---|---|---|
| Biomass allometric equation | According to the biomass of different diameter at breast height (DBH) of tree species, allometric growth of biomass was established through the relationship between the measured data and DBH and tree height. | DBH, tree height, biomass allometric equation parameters of corresponding tree species. | High precision in small scale studies. | The acquisition of parameters is time- and labor-consuming, so it is not suitable for large-scale studies. | [28] |
| Biomass expansion factor | The measured DBH and height of the trees in the plot were used to obtain the individual tree volume, and then, the plant biomass was calculated using the trunk volume density and biomass expansion factor. | Tree species, number, volume of tree species, wood density and biomass expansion factor. | Suitable for estimating the carbon storage in large areas of forest. | The parameters are complicated, difficult to obtain and low precision in small scale. | [25] |

**Table 1.** *Cont.*

| Methods | Theories | Required Data | Advantage | Disadvantage | Reference |
|---|---|---|---|---|---|
| Photosynthetic rate | According to the carbon sequestration rate of leaf area of different vegetation, the leaf area of vegetation was measured, so as to estimate the carbon storage of green space. | Leaf area index, net carbon sequestration rate per unit leaf area, photosynthetic time of different plants. | High precision in small scale studies. | It is difficult to obtain the parameters and is not suitable for large-scale research. | [29] |
| Carbon density | The organic carbon storage of UGS is estimated by using carbon density and area of green space. | Average carbon density of UGS, area of UGS. | Suitable for the national scale and built environment, fast and easy to operate. | The accuracy of the data has great influence on the results and cannot reflect the difference of green space properties. | [30] |
| Greenhouse gas inventory | According to the accumulation of vegetation canopy area and carbon storage per unit area, the total carbon storage of green space was calculated. | Vegetation canopy area, carbon storage per unit area of vegetation/green space. | Suitable for rapid measurement of carbon storage. | It is difficult to obtain urban measurement data. | [31] |
| City-Green model | Based on Geographic Information System (GIS) and remote sensing technology, the ecological benefit evaluation model of green space based on ArcGIS platform was developed. | Tree leaf density, tree height growth rate, DBH growth rate, crown shape, leaf shedding, height of the largest tree, etc. | Suitable for large-scale green space carbon storage research. | Most of the model parameters come from the United States, and the applicability of the parameters in other cities needs to be verified. | [32] |
| i-Tree model | The annual benefit structure, function and value of trees were quantified according to the inventory data of UGS. | Field data, local hourly air pollution and meteorological data. | Higher precision on a small scale in urban area. | Applicability in other cities has yet to be verified. | [33] |
| Pathfinder system | A carbon calculation system for urban landscapes based on landscape materials, carbon sink pathways and carbon cost parameters. | Type and quantity of landscape materials, carbon sink of vegetation, carbon cost. | Simple operation. | It is difficult to determine parameters and obtain data. | [34] |
| Remote sensing inversion | A variety of vegetation indices were selected to obtain the optimal relationship between different land cover types and construct the integrated model. | Biomass obtained from remote sensing image and sample inventory. | Suitable for the carbon storage estimation in large-scale UGS. | The spatial resolution of remote sensing images is required to be higher in the urban area. | [35] |

i-Tree model is powerful and more targeted, which can accurately calculate various ecosystem service benefits generated by urban green space and has been widely used and highly recognized by the academic community in Europe and America [33]. In this study, the downtown area of Kaifeng is taken as the study area. On the basis of the field survey, localization and verification of the i-Tree model, we precisely estimate the UGS carbon storage by combining remote sensing data. The appropriate scale for spatial mapping of UGS carbon storage was analyzed too.

## 2. Materials and Methods

### 2.1. Study Area

As a city in central China, it is representative in the construction of ecological landscape. We chose Kaifeng, Henan Province, China, which is close to us, as the study area. Kaifeng is located at $34°11'45''\sim35°01'20''$N, $113°52'15''\sim115°15'42''$E, with a total area of 6239 km$^2$. The terrain shows a general trend of slight inclination from northwest to southeast, with an altitude of 69~78 m. Due to the influence of monsoons, it shows typical climate characteristics of a warm temperate continental monsoon, with four distinct seasons. The average annual temperature is 14.52 °C and the average precipitation is 627.5 mm. This study

area is located in the downtown area of Kaifeng (34°45′N~34°50′N, 114°13′E~114°23′E), with a total area of 101.24 km² (Figure 1). Based on the global 1 km spatial distribution of population data provided by WorldPop, we obtained the total population of the study area as about 652,200 [36].

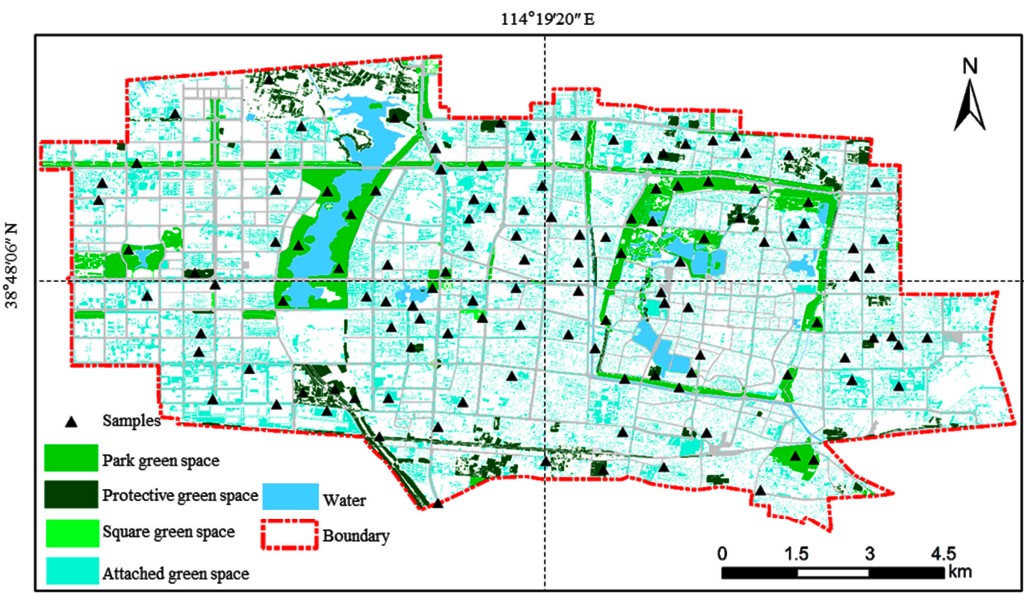

**Figure 1.** Study area.

*2.2. Data*

2.2.1. High-Resolution RGB Satellite Image

Google Earth (https://earth.google.com/web/, accessed on 11 April 2023) is a virtual globe software developed by Google. Users can browse high-resolution RGB satellite images around the world for free through the client, and the resolution can reach less than 0.5 m. In this study, RGB sub-meter images were collected on Google Earth for spatial information extraction and type division of UGS.

2.2.2. Sentinel-2A/B Image

The multispectral remote sensing images were selected from Sentinel-2A/B with 10 m spatial resolution published by the European Space Agency (https://scihub.copernicus.eu, accessed on 11 April 2023) in 2022 [37]. Sentinel-2A/B images have 13 bands, which can support the calculation of three vegetation indices (Normalized Vegetation Index, NDVI; Ratio Vegetation Index, RVI; Difference Vegetation Index, DVI) and provide important data support for the next study.

*2.3. Methods*

2.3.1. Extraction and Classification of Urban Land Type

Firstly, we used high-resolution RGB images from Google Earth to visually interpret and extract UGS in the downtown area of Kaifeng. The high-precision spatial data of UGS can provide support for the accurate measurement of UGS carbon storage. Sub-meter satellite images can also reduce the estimation error caused by the loss of small green space patches as much as possible. Secondly, the neighborhood blocks within the downtown area of Kaifeng were divided. Then, combined with map and field survey data for attribute classification, we finally obtained the land use data and detailed UGS type data by integrating land use types and UGS spatially (Figures S2 and S3).

### 2.3.2. UGS Sample Survey

In this study, 123 survey samples were selected according to the land use type and vegetation cover of the green space observed by high-resolution remote sensing images from Google Earth (Figure 1). We randomly investigated the trees and shrubs. The main way to obtain vegetation attributes is by manually measuring and recording the growth parameters of vegetation in detail. These survey samples were set to 10 m × 10 m. For green space of a sample with a length and width of less than 10 m, the entire plot was selected. Random trees and shrubs were selected in each sample to record the plant growth parameters, including the species, DBH, crown width, height, crown missing, and dieback rate of trees, the species, branch diameter, crown width, height, and other attributes of shrubs.

### 2.3.3. Calculation of UGS Carbon Storage

In urban green space, lawns contribute little to total carbon storage [38], and most of the carbon absorbed by lawns will be released back into the atmosphere due to regular mowing or winter wilting too [20]. Therefore, this study only considers trees and shrubs in UGS. The calculation formula for total carbon storage is as follows:

$$S = S_a + S_s \tag{1}$$

where $S$ is the total green space carbon storage (kg), $S_a$ is the tree carbon storage (kg), and $S_s$ is the shrub carbon storage (kg).

(i)     Tree carbon storage

$$S_a = \sum_{i=1}^{n} \left[ \alpha \delta_i \left( m_{ai} + m_{gi} \right) \right] \tag{2}$$

where $\alpha$ is the carbon content of biomass, using a value of 0.5 [39]; $\delta_i$ is the biomass correction coefficient of the $i$-th tree, which is assigned according to the health status of the plant, and its value is derived from the health assignment matrix of tree (Table S1); $m_a$ is the above-ground biomass (kg), which is obtained according to the mixed allometric model constructed in Table A1; $m_g$ is the under-ground biomass (kg), which is obtained by $m_g = m_a \beta$; and $\beta$ is the root-shoot ratio, which is 0.26 [40].

(ii)     Shrub carbon storage

$$S_s = \sum_{k=1}^{m} \left( 1.249 \alpha P_k A_k \right) \tag{3}$$

where $P_k$ is the shrub cover of the $k$-th sample measured (%); $A_k$ is the area of the $k$-th sample (m$^2$).

### 2.3.4. i-Tree Eco Model

The i-Tree model has become a classic software to quantify urban forest structure and ecological functions (www.itreetools.org, accessed on 2 March 2023). The i-Tree model has many modules, and this study utilized the i-Tree Eco (Urban Forest Effects Model, UFORE) to calculate carbon storage of UGS. UFORE is designed to use field data and local hourly meteorological, as well as air pollution, data to give a detailed characterization of ecological services, including carbon sink and storage of UGS [41]. The UFORE model has been widely used in case studies across the world [42,43]. This study used pre-stratified random sample plots for UFORE to assess the carbon storage of UGS in Kaifeng downtown. Local hourly meteorological data and pollution data have been uploaded to the i-Tree team and passed by the review. The verification results of UFORE in this study also showed that it can support our study (Figure S1).

2.3.5. Vegetation Index and Remote Sensing Mapping of Carbon Storage

The sample survey data provided the input data of the i-Tree Eco model. Further, remote sensing mapping was carried out through the vegetation index to estimate the overall carbon storage of the study area. In this study, we selected three common vegetation indices on the basis of Sentinel-2A/B images and combined them with the calculation results of i-Tree Eco to construct a regression model of UGS carbon storage. The vegetation index and calculation formula are shown in Table 2.

**Table 2.** Vegetation index and its formula.

| Vegetation Index | Formula |
|---|---|
| NDVI | $NDVI = \frac{NIR - R}{NIR + R}$ |
| RVI | $RVI = \frac{NIR}{R}$ |
| DVI | $DVI = NIR - R$ |

Note: *NIR* represents the near-infrared band and *R* represents the visible red band.

The spatial estimation of UGS carbon storage was established by a regression model fitted between vegetation index and sample survey data. The three vegetation indices were used, respectively, and the regression model with the best fitted line was selected. At the same time, in order to test the accuracy of the estimation model, the 123 sample survey data were divided into two parts, namely, 100 sample data were used to establish the regression model, and the remaining 23 sample data were used as the test data set to evaluate the model accuracy. Furthermore, in order to obtain the minimum resolution of remote sensing data that can be used in the estimation of urban carbon storage in large regions, i.e., the minimum spatial scale, six different grid scales were set up to explore the scale effect on accuracy of UGS carbon storage estimation.

## 3. Results

### 3.1. Spatial Distribution of UGS

According to the land use type and the spatial locations of green spaces, 17 types of UGS were classified (Figure 2 and Figure S4). In order to facilitate the analysis, these 17 UGS types were summarized into park green space, protective green space, square green space, and attached green space. The attached green space includes residential area green space, commercial/financial green space, educational/scientific green space, road green space, industrial green space, administrative office green space, and other green spaces, which included eight categories too. The basic pattern of the park green space presented a dual-center distribution of east and west parts in space. Protective green space mostly distributed on the edge of the city, which was relatively concentrated in the northwest and southwest. Square green space was less distributed in the study area. As an important UGS type, attached green space was widely distributed in urban areas. Among them, residential green space was widely distributed in all orientations of the city, and the distribution of commercial/financial green space was highly fragmented. Most of the educational/scientific green space was distributed in the north and northwest of the study area. Road green space was distributed near to the roads. Industrial green space was concentrated in the southwest and southeast of the city. Administrative office green space only had a small, concentrated distribution in the central area, and other green space was mainly distributed more in the west of the city.

The total area of UGS in the study area was 26.41 km$^2$, and the proportion of various types of green space has a large difference (Table 3). The total area of residential green space was 7.29 km$^2$, accounting for 27.4% of the entire study area. Followed by the park green space, the area was 5.87 km$^2$, accounting for 22.23%. The area of protective green space was 3.05 km$^2$, accounting for more than 10%. The other types of green space accounted for less than 10% of the entire study area, and the area order from large to small was other green space > industrial green space > educational/scientific green space > road green space > commercial/financial green space > administrative office green space > square green

space. The overall greening rate of the study area was 26.09%. Park green space, protective green space, and square green space had the highest greening rate, which was 75.84%, 51.09%, and 41.38%, respectively. Among the attached green spaces, the greening rates of educational/scientific green space and administrative office green space were 28.67% and 27.07%, respectively, and they were both higher than the average greening rate of the study area. The total area of green space per capita in the study area is 40.49 $m^2$, of which the per capita residential area green space is the largest, reaching 11.18 $m^2$, and the per capita Square green space is the smallest, at less than 1 $m^2$.

**Table 3.** Land use type and area of UGS.

| | Land Use Type | Area of Green Space (km$^2$) | Area (km$^2$) | Greening Rate (%) | Per Capita Green Space (m$^2$) |
|---|---|---|---|---|---|
| | Park green space (G1) | 5.87 | 7.74 | 75.84% | 9.00 |
| | Protective green space (G2) | 3.05 | 5.97 | 51.09% | 4.68 |
| | Square green space (G3) | 0.12 | 0.29 | 41.38% | 0.18 |
| | Educational/scientific green space (A3) | 2.15 | 7.50 | 28.67% | 3.30 |
| | Administrative office green space (A1) | 0.36 | 1.33 | 27.07% | 0.55 |
| Attached green space (XG) | Industrial green space(M) | 2.26 | 8.92 | 25.34% | 3.47 |
| | Residential area green space (R) | 7.29 | 35.08 | 20.78% | 11.18 |
| | Other green space (O) | 2.56 | 13.15 | 19.47% | 3.93 |
| | Road green space (S) | 1.46 | 9.20 | 15.87% | 2.24 |
| | Commercial/financial green space (B) | 1.29 | 8.41 | 15.34% | 1.98 |
| | Water (L) | — | 3.64 | — | — |
| | Total | 26.41 | 101.24 | 26.09% | 40.49 |

Note: The UGS area per capita is calculated by areas of various-type and total green spaces/total population, respectively.

### 3.2. Carbon Storage of UGS in Sample

The composition of UGS produced various community structure modes with the combination of trees, shrubs, and herbs, which reflect the complexity of UGS. In this study, the community structure of UGS was divided into five modes: tree–shrub–herbs, tree–shrubs, tree–herbs, shrub–herbs, and tree community structures. Among them, the tree–shrub–herbs mode accounted for the highest proportion in all samples, reaching 51.22%. The shrub-herbs mode accounted for the smallest proportion in all samples, only 0.81% (Figure 3a). The results of the sample survey also showed that the vegetation allocation of UGS in the study area was more dominated by trees. Almost all UGS had trees, while shrubs were relatively small, and individual lawns with a certain scale were even harder to find. The average carbon density in the sample was 5.89 kg m$^{-2}$. Among them, the average carbon density of trees in various types of UGS was between 3.96 and 7.41 kg m$^{-2}$, and the carbon density of shrubs was between 0.02 and 0.1 kg m$^{-2}$. There were differences in the carbon density of various green space types, among which park green space, protective green space, and square green space, as the main bodies of urban carbon storage, had the highest carbon density among all UGS types. The average carbon density of the trees showed the order characteristics of attached green space < square green space < protective green space < park green space, while the highest carbon density distribution of shrubs was in square green space (Figure 3b).

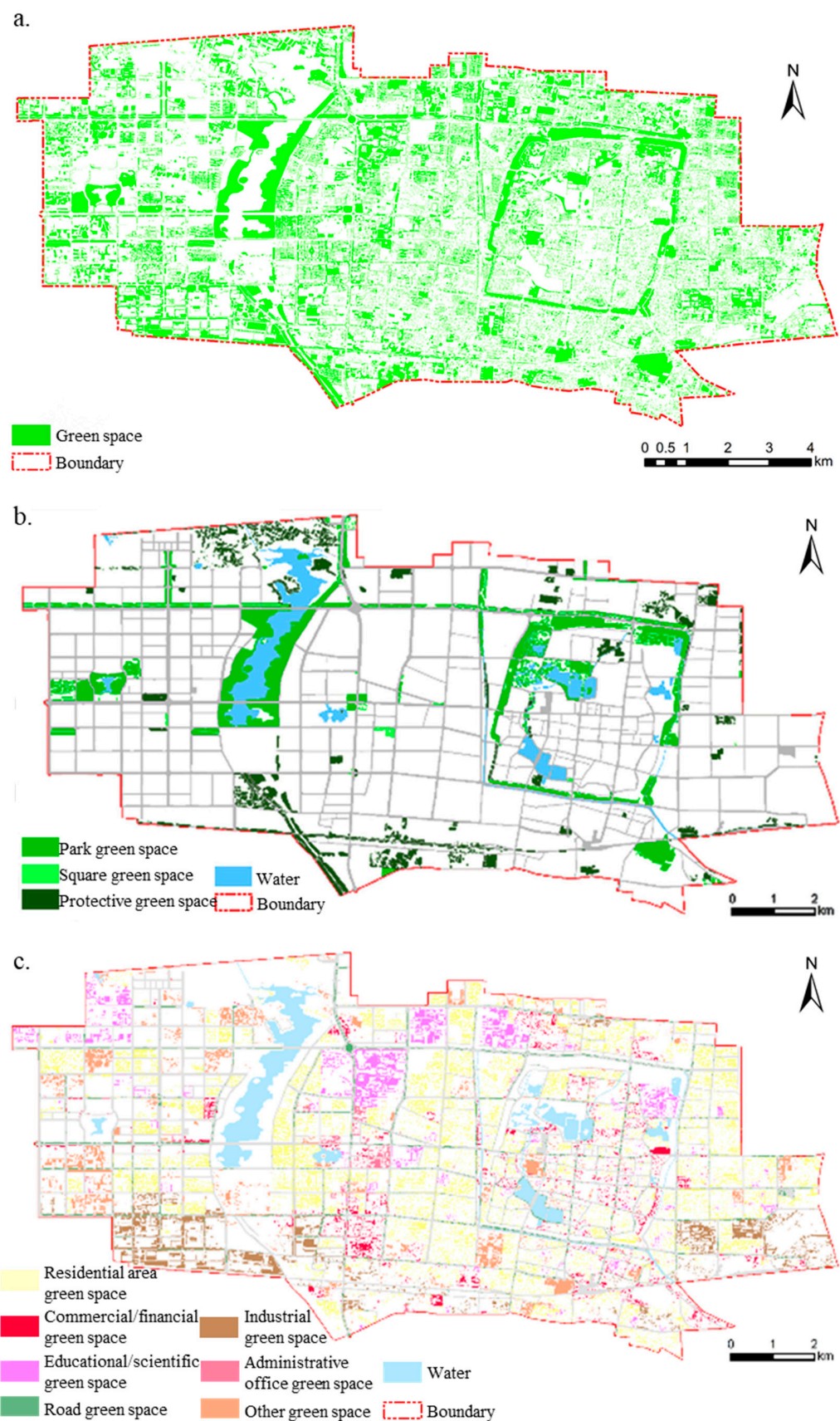

**Figure 2.** Spatial distribution of UGS. (**a**) Overall spatial distribution; (**b**) spatial distribution of three types of UGS; (**c**) spatial distribution of attached green space.

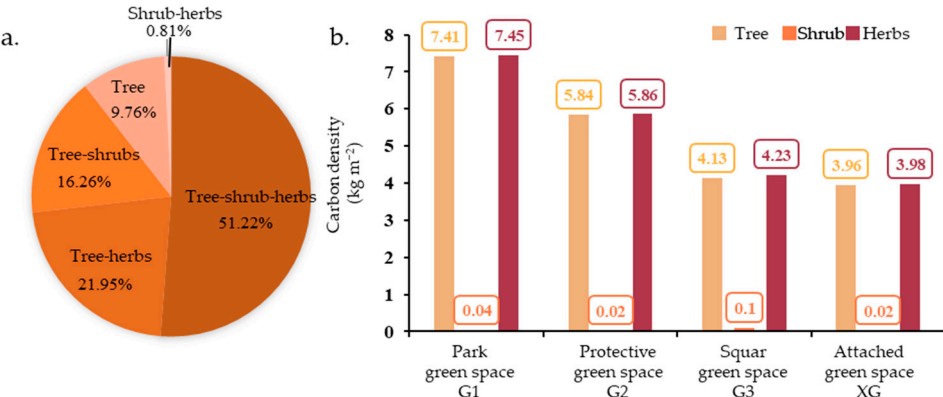

**Figure 3.** The proportion of various plant community structure modes (**a**) and average carbon density of various UGS (**b**).

### 3.3. Estimation and Mapping of UGS Carbon Storage

3.3.1. Estimation of UGS Carbon Storage

By analyzing the statistical relationship between different vegetation indices and above-ground carbon storage in 100 samples and considering the correlation and fitting effect of the model, the regression model with the best fitting equation was finally chosen (see Table A2 for details). A cubic curve model ($Y = -126.827 + 2468.4 NDVI - 6555.944 NDVI^2 + 6945.717 NDVI^3$) had the highest decision coefficient ($R^2 = 0.827$). We thus used the cubic curve model to estimate UGS carbon storage in the study area. Then, the remaining 23 sample data were used to estimate the model accuracy. The results showed that the regression model had high accuracy and confidence (Table S2).

Combined with the spatial distribution of UGS, the spatial distribution map of UGS carbon storage in the study area was obtained (Figure 4). The total carbon storage was 114,389.17 t, and the per capita carbon storage was 175.39 kg. The average carbon density was 4.33 kg m$^{-2}$. The types of UGS with carbon storage accounting for more than 10% were park green space, residential green space, protective green space, and the other green spaces (G1, G2, R, O), accounting for 73.37% of the total carbon storage. Among them, the park green space carbon storage was the highest, reaching 30,312.44 t, accounting for 26.50% of the total carbon storage. This was followed by residential green space and protective green space, reaching 21,400.82 t and 19,939.40 t, accounting for 18.71% and 17.40%, respectively. The other green spaces carbon storage was 12,299.48 t, accounting for 10.75%. Among the remaining UGS types, the carbon storage proportions of only two green space types were more than 5%, namely, educational/scientific green space and industrial green space, that were 9.06% and 9.09%, respectively (Figure 5, Table S3).

There were large differences in the average carbon storage and carbon density among various UGS types. Green space carbon storage fluctuated greatly, ranging from 0.34 t to 4017.67 t (Tables S3 and S4). The carbon density of protective green space was the highest among various UGS, which was 6.52 kg m$^{-2}$, followed by square green space and park green space, which were 5.70 kg m$^{-2}$ and 5.16 kg m$^{-2}$, respectively. The reasons causing the large differences may be that cities tend to design green spaces with a large area during the initial planning and building period, and some small green spaces will be built in the later stages of urban development as a supplement to urban greening. Furthermore, different greening management measures and maintenance standards will also cause the differences in carbon storage.

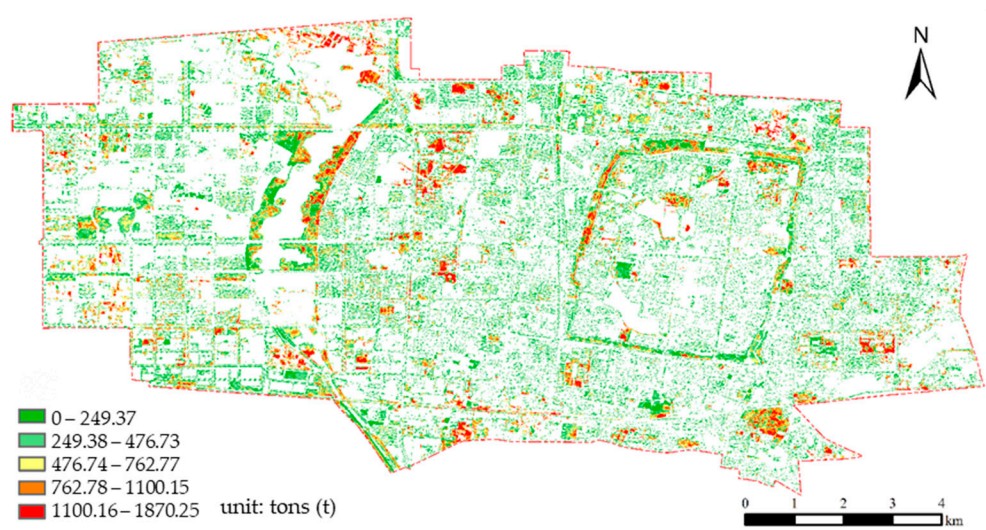

**Figure 4.** Spatial distribution of UGS carbon storage.

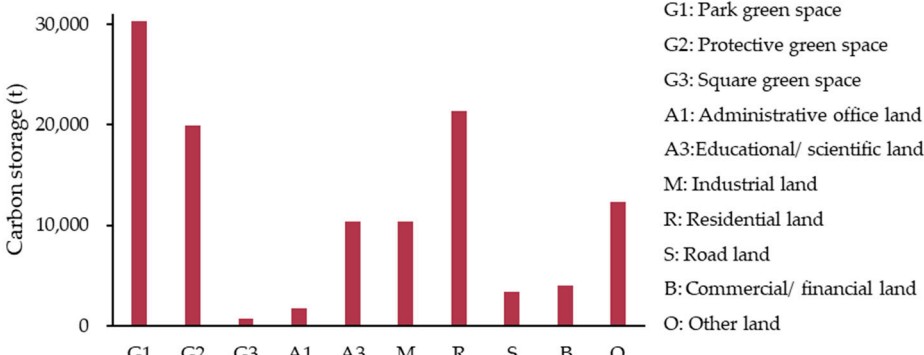

**Figure 5.** Carbon storage of various UGS types.

### 3.3.2. Appropriate Spatial Mapping Scale Analysis

The appropriate spatial mapping scale is particularly important for large-scale or regional assessment of urban carbon storage. In this study, six different grid scales were set up to explore the scale impact on urban carbon storage estimation accuracy. The results showed that the standard deviation of carbon storage decreased along with the decrease in the grid scale, indicating that the scale is an important factor in estimating urban carbon storage (Table 4). When the spatial resolution was less than 1 km, the minimum value of grid carbon storage is 0 t, showing that the urban carbon storage can be effectively distinguished in space only when the spatial resolution is less than 1 km. That is, a 1 km or more scale actually cannot be used for mapping of urban carbon storage, as it will overlook a large number of scattered UGS. In terms of the average value and standard deviation of urban carbon storage, the average value and standard deviation of urban carbon storage had an inflection point at the 0.25 km scale, and the average value and standard deviation increased exponentially after exceeding the inflection point of 0.25 km. Therefore, taking into account the data source, data operation amount and cartographic accuracy, the spatial resolution of 0.25 km is the minimum scale requirement for spatial mapping of urban carbon storage (Figure 6). To a certain extent, the above analysis also indicates the importance of high-precision mapping data to estimate the urban carbon storage.

**Table 4.** Statistic values of urban carbon storage at different scales.

| Scales | n | Minimum (t) | Maximum (t) | Average (t) |
| --- | --- | --- | --- | --- |
| 0.1 km × 0.1 km | 10,685 | 0 | 126.39 | 10.71 |
| 0.25 km × 0.25 km | 1735 | 0 | 482.07 | 65.93 |
| 0.5 km × 0.5 km | 469 | 0 | 1529.64 | 243.91 |
| 1 km × 1 km | 135 | 2.54 | 2371.07 | 847.34 |
| 2 km × 2 km | 40 | 10.69 | 7011.80 | 2859.80 |
| 4 km × 4 km | 13 | 30.68 | 22,966.20 | 8799.35 |

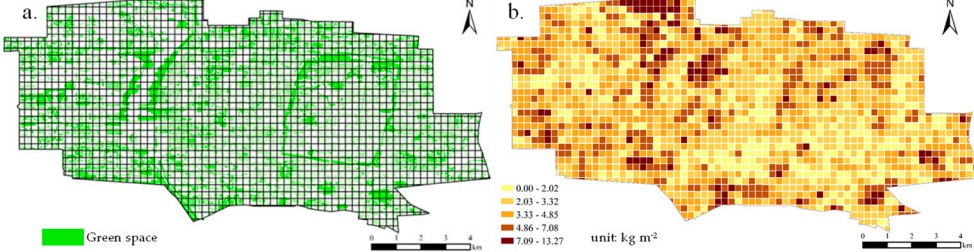

**Figure 6.** Spatial distribution of UGS (**a**) and urban carbon density (**b**) at 0.25 km × 0.25 km grid scale.

## 4. Discussion

The UGS carbon storage shows that cities are far from just cement, and the construction of eco-cities has made urban areas play an important role in carbon sequestration [44]. In Henan Province, where Kaifeng is located, the average green space carbon density of the downtown area of Kaifeng is similar to that of 4.43 kg m$^{-2}$ in Luohe [45] and lower than that of 5.48 kg m$^{-2}$ in Zhengzhou [46]. Additionally, trees are the major contributor to the carbon pool of UGS, and the conclusion of this study is consistent with others [47]. Nowak et al. [48] conducted a field survey on the carbon storage of green spaces in 28 cities in the United States, showing that the carbon storage density of UGS trees could reach 7.69 kg m$^{-2}$, and the total carbon storage reached 600 million tons. Using high-resolution remote sensing data, Pasher et al. [49] conducted a study on the carbon sequestration benefits of UGS in Canadian cities, and the results also showed that the carbon storage of urban trees in Canada reached 34 million tons. Our previous study also showed that the carbon sequestration within 79.3% of the global urban areas increased year to year [50]. All of the above studies showed that urban carbon storage contributes to important impacts on the regional ecology and climate. Therefore, the ecological and climatic effects of urban vegetation should not be ignored or considered negligible in the carbon cycle [51], because urban vegetation not only directly affects regional ecology and climate by biogeophysical mechanisms but also indirectly affects the regional environment by carbon sequestration of biogeochemical cycle [44,52].

In this study, we carried out the parameter localization of the i-Tree model and verified the results, so as to carry out the corresponding simulation and analysis. Nevertheless, the development of the i-Tree model is based on the climatic conditions and economic level of local cities in the United States, and some tree species in Kaifeng may not be included in the model database. So, in the selection of tree species, we try to maintain the consistency between the tree species types in the study area and the model itself, but it is bound to reduce the reliability of the analysis conclusions. At the same time, the i-Tree model requires a series of assumptions to estimate urban green space carbon storage, such as unmeasured root–shoot ratios, non-city-specific growth rates and adjustments for tree conditions, light and land use, as well as decomposition rates [53]. These limitations may result in a high or low estimation. Different models may cause some differences in simulation results, but it does not affect the main focus of this study—the scale effect.

In this study, we used three vegetation indices of NDVI, RVI and DVI to fit the 11 formulas. The results showed that all of the fitting formulas passed the significance test,

while the formula with the highest fit included only one vegetation index, NDVI. This is mainly due to the high correlation of the three vegetation indices. The maximum NDVI data of NASA GIMMS is even just a function of RVI [54]. Therefore, the fitting formula composed of three highly correlated vegetation index variables also has high collinearity. Collinearity, on the other hand, can lead to distorted or less stable model estimates. Although this study uses the formula of the maximum fitting coefficient to maintain accuracy as much as possible, it still cannot solve the problem of reduced estimation accuracy or stability caused by collinearity. Therefore, when completing the scale expansion study, the problems of specific study areas still need to be analyzed in detail.

Nevertheless, it is still difficult for current studies to break through multiple cities. Exploring the suitable or minimum spatial mapping scale of urban carbon storage is a prerequisite for both accurate urban carbon storage estimation and scale extension of study area, as well as urban and regional climate effect simulation. Although remote sensing has been introduced in many studies, the issue of mapping scale is still limited by the spatial resolution of the multi-source and raw remote sensing data, resulting in the lack of analyzing the suitable mapping scale [26]. The high-resolution remote sensing image can directly obtain high precision for urban carbon storage, but the data resource will directly affect the study scale and computational efficiency [55,56]. The scale of 0.25 km obtained in this study is an unexpected result, which can be used as a reference value for urban carbon storage mapping and regional climate modeling.

It is of great significance to study the influencing factors of UGS carbon storage for urban low-carbon development [44]. From the perspective of urban management, appropriately increasing the planting density of trees and incorporating more suitable trees into urban areas can effectively increase the carbon sequestration of UGS [57]. Previous studies have shown that the vegetation structure composition has a greater impact on the carbon sequestration capacity of UGS and also confirmed that the impacts of the increase in green space carbon sequestration capacity is greater than the increase in green space area [58–60]. Therefore, further studies should combine the principles of ecology and the landscape index of UGS, to analyze the impacts of the tree–shrub–herbs composite structure or even the three-dimensional greening on the UGS carbon storage, so as to provide corresponding suggestions for enabling UGS to exert greater urban ecosystem service functions [61,62].

## 5. Conclusions

In view of the lack of refined carbon storage estimation and appropriate spatial mapping scale of UGS, this study used field survey data and high-resolution remote sensing images and combined them with the i-Tree Eco model to classify the types of UGS in the downtown area of Kaifeng. A cubic curve regression model was constructed for estimating spatial urban carbon storage, and the minimum scale for spatial mapping of urban carbon storage was obtained too. The results showed that the total area of UGS in the study area was 26.41 km$^2$, and the total urban carbon storage was 114,389.17 t. The per capita green space area and per capita carbon storage were 40.49 m$^2$ and 175.39 kg, respectively. The average carbon density of protective green space, square green space, and park green space ranked in the top three among all UGS types, with 6.52 kg m$^{-2}$, 5.70 kg m$^{-2}$ and 5.16 kg m$^{-2}$, respectively. The urban carbon storage had obvious scale dependence, and the spatial carbon storage showed different characteristics at different scales. The minimum spatial resolution requirement for urban carbon storage mapping was 0.25 km or less.

Our study results confirm that the urban area is not a concrete jungle anymore but has considerable carbon storage, which makes up for the shortcomings of the current research system on urban carbon storage issues to a certain extent. This study not only provides a strong reference for assessing urban carbon storage and modifying regional climate models, but it also has important guiding significance for the formulation of urban low-carbon development strategies and the spatial optimization of UGS.

**Supplementary Materials:** The following supporting information can be downloaded at: https://www.mdpi.com/article/10.3390/rs16020217/s1, Figure S1: Comparison of carbon storage between i-Tree model and allometric equations. The top 10 tree species with total carbon storage were selected; Figure S2: Land use types in the downtown area of Kaifeng city; Figure S3: The area proportion of various land use type; Figure S4: Spatial distribution of 3 types of green space and 7 types of attached green space; Table S1: Tree biomass correction coefficient assignment matrix; Table S2: Regression model validation index; Table S3: Carbon storage of various types of urban green space; Table S4: Carbon density of various types of urban green space.

**Author Contributions:** Conceptualization, N.L., Y.C. and L.D.; methodology, N.L., Y.C. and L.D.; validation, L.D.; formal analysis, N.L. and L.D.; data curation, G.Y.; writing—original draft preparation, N.L., Y.C., L.D. and M.C.; writing—review and editing, N.L. and Y.C. All authors have read and agreed to the published version of the manuscript.

**Funding:** This research was funded by the National Natural Science Foundation of China, grant number 42071415; Xinyang Institute of Ecology 2023 Open Fund, grant number 2023XYMS014; Central Plains Young Top-notch Talent Program and Excellent Textbook Project for Graduate Students in Henan Province, grant number YJS2023JC22.

**Data Availability Statement:** High-resolution RGB satellite images in this study are provided by Google Earth (https://earth.google.com/web/, accessed on 11 April 2023). The multispectral remote sensing images are from Sentinel-2A/B published by the European Space Agency (https://scihub.copernicus.eu, accessed on 11 April 2023). The global 1 km spatial distribution of population data are provided by WorldPop (https://hub.worldpop.org/geodata/summary?id=24777, accessed on 24 December 2023).

**Conflicts of Interest:** The authors declare no conflict of interest.

## Appendix A

**Table A1.** Biomass allometric equation of various tree species. The top 10 tree species with total carbon storage were selected.

| Tree Species | Allometric Equation | Sources |
|:---:|:---:|:---:|
| Koelreuteria paniculata | $B_t = 0.10994D^{2.48438}$ | [63] |
| Cinnamomum camphora | $B_t = 0.10378D^{2.53500}$ | [63] |
| Populus tomentosa | $B_t = 0.0262D^{2.9404}$ | [64] |
| Salix babylonica | $B_B = 0.0326(D^2H)^{0.8472}$ <br> $B_L = 0.0250(D^2H)^{1.1778}$ | [65] |
| Platanus acerifolia | $\lg(B_t) = -1.161443 + 0.913291 \times \lg(D^2H)$ | [66] |
| Ailanthus altissima | $B_t = 0.8760 + 0.0124(D^2H)$ | [67] |
| Eucommia ulmoides | $\ln(B_t) = 0.8007 \times \ln(D^2H) - 0.8114$ | [68] |
| Ligustrum lucidum | $B_t = 0.13999D^{2.34273}$ | [69] |
| Fraxinus chinensis | $B_t = 2.1893 + 0.032949(D_{1.3}^2H)$ | [70] |
| Platycladus orientalis | $B_B = 0.1642D^{1.8804}$ <br> $B_L = 0.3222D^{1.4818}$ | [71] |

**Table A2.** Regression models and statistic parameters.

| Regression Model | Equation | $R^2$ | Adjusted $R^2$ | F | Sig. |
|:---|:---:|:---:|:---:|:---:|:---:|
| Multiple linear regression | $Y = -180.012 + 352.596X_1 + 103.614X_2 + 0.43X_3$ | 0.816 | 0.810 | 142.008 | 0.00 |
| Stepwise linear regression | $Y = -177.333 + 481.497X_1 + 105.631X_2$ | 0.815 | 0.811 | 213.895 | 0.00 |
| Linear regression | $Y = -616.771 + 2137.063X_1$ | 0.686 | 0.682 | 213.695 | 0.00 |
| Quadric model | $Y = 431.466 - 2268.733X_1 + 4038.485X_1^2$ | 0.811 | 0.808 | 208.744 | 0.00 |

**Table A2.** *Cont.*

| Regression Model | Equation | $R^2$ | Adjusted $R^2$ | F | Sig. |
|---|---|---|---|---|---|
| Cubic curve model | $Y = -126.827 + 2468.4X_1 - 6555.944X_1{}^2 + 6945.717X_1{}^3$ | 0.827 | 0.822 | 153.169 | 0.00 |
| Composite model | $Y = 48.184 \times 50.910^{X_1}$ | 0.820 | 0.819 | 447.787 | 0.00 |
| S model | $Y = e^{7.024 - 0.356/X_1}$ | 0.436 | 0.431 | 75.856 | 0.00 |
| Power model | $Y = 1321.642 \times X_1{}^{1.600}$ | 0.706 | 0.703 | 235.803 | 0.00 |
| Growth model | $Y = e^{3.875 + 3.930X_1}$ | 0.820 | 0.819 | 447.787 | 0.00 |
| Exponential model | $Y = 48.184e^{3.930X_1}$ | 0.820 | 0.819 | 447.787 | 0.00 |
| Logistic model | $Y = 1/(0 + 0.021 \times 0.0196^{X_1})$ | 0.820 | 0.819 | 447.787 | 0.00 |

Note: $X_1$ is NDVI; $X_2$ is RVI; $X_3$ is DVI.

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
