# Peer review of "Estimation for Refined Carbon Storage of Urban Green Space and Minimum Spatial Mapping Scale in a Plain City of China"

_remotesensing, doi:10.3390/rs16020217_

Round 1

Reviewer 1 Report

Comments and Suggestions for Authors

Major concern:

In the conclusion,  it is said 'this study used survey sample data remote sensing image' but this is not crearly explained in l.129. If those survey samples mentioned in l.129 are from remote sensing, that should be clarified at this point, and it should be specified from which remote sensing source, since Google Earth and Sentinel are mentioned.

Something similar happens in l.120: it should be indicated if RGB images are from Google Earth or Sentinel.

Moreover, some explanation should be included in the methodology followed regarding grid scales setting mentioned in l.283

Minor revisions:

l.12 'cities' instead of 'Cities'

l.55 revise 'etc.', maybe other coefficients or parameters should be mentioned

l.62 revise if it is 'methods' or 'models'

l.66. check the english grammar in 'these are obvious differences spatially'

l. 73 check letter type and size in the name 'Tang'

l. 78 revise the sentence 'analysis of combined with remote sensing': What is combined?

l. 81 revise 'of in'

Table 1. revise each grid cell of the table: start with capital letter and finish with '.', as made in the majority of the grid cells of the table

l.92. check the number '2' that it is supposed to be a superindex

l.95 check if there is too many spaces between '14.52' and 'ºC'

l. 102 - 106 remove these lines since they are repeted from l. 91-95

l. 115 rewrite the sentence 'spatial resolution of 10 in July': the spatial resolution is the same always, not only in July

l. 146 'where' instead of 'Where'

l. 149 - 151 in the names 'ma' and 'mg' 'a' and 'g' should be subindex

Figure 2. In Figure 3 it is written (a), so the caption of Figure 2 should be written also (a) and (b) instead of a. and b.

l.207 In this line it is said '26.41 km2' but in l.97 it is written 101.24 km2. This issue should be clarified, particularly if it is referred to the same area or not

l. 266 check spelling 

l. 309 'kg' instead of 'km'

l.315 the units for the values regarding carbon storage are indicated as kg m-2 in all the manuscript with the exception of this line, please, revise it

l. 351 the value '26.43 km2' only appears in the Conclusions and Abstract but not in the Results, please, revise it

Reviewer 2 Report

Comments and Suggestions for Authors

The research aims to study a method for carbon storage estimation in a city in China. This method is useful for green infrastructure managers and urban planners. The paper can be accepted after minor improvements. 

In line 137, is mentioned fewer lawns in Kaifeng. Fewer than what? Probably, the authors want to say that lawns are few and are excluded from this study.

Lawns and other herbaceous vegetation stare carbon in urban areas even if they are mowed. Zero carbon storage is in bare land. A football field is one ha and has a specific volume of biomass, thus it should be included in equation 1 in line 141. In google maps, I could visualize several lawns and herbaceous vegetation between trees and shrubs. 

The i-tree has mostly tree species from the USA. All trees of the Chinese city were included in i-tree? What did the authors do for trees that are not included in the i-tree database?

The tree species in Table A1 are the most common in this city or the best in carbon storage according to i-tree?

In Table 2, it is A2 and not A3, because in line 65 it is mentioned Figure 5 showing A2 and M, but not A3. 

The letters in Figure 5 should be explained in the legend in line 269 because each figure should be able to stand alone without searching in the text. 

In line 309, it is kg and not km.

The authors use the latin word "arbor" instead of tree. This is not commonly used and we usually use "arbor" and "arboriculture" when we treat individual trees. In urban planning if green infrastructure I suggest to use the term treeand "urban forestry" because it is a study of a system and not individual trees. 

In chapter 2.1 please mention the population of the study area. If it is 1.7 million people you can add a new column in Table 3 showing the green areas per person. 

Reviewer 3 Report

Comments and Suggestions for Authors

I am pleased to review the manuscript entitled “Estimation for refined carbon storage of urban green space and minimum spatial mapping scale in a plain city of China”. This study takes the urban region of Kaifeng as study area, combining the I-Tree ecological model with remote sensing data to construct an empirical model for estimating the UGS carbon storage, which has scientific significance. I have some issues to discuss with the authors.

1.     In this study, the author only provided one empirical model for estimating UGS carbon storage. I believe that the author should at least divide UGS into two categories: trees and shrubs, and construct an empirical model for estimating carbon storage based on vegetation index respectively.

2.     Before constructing the UGS carbon storage estimation model, collinearity diagnosis of NDVI, RVI, and DVI factors should be conducted.

3.     I did not find Table S2 and Table S3 in the manuscript.

4.     I strongly suggest that this study should provide rigorous verification of the accuracy of carbon storage estimation results.

5.     Figure 4 lacks units.

6.     The English writing of the manuscript needs improvement.

Comments on the Quality of English Language

The English writing of the manuscript needs improvement.

Reviewer 4 Report

Comments and Suggestions for Authors

Li et al. estimated the spatial carbon storage of UGS in the downtown area of Kaifeng city, using remote sensing dataset and the i-Tree model. Overall, the article is well-written, but there are still some parts that are unclear or not rigorous enough. The detailed comments are as the followings:

1.Please provide an explanation in the main text regarding why Kaifeng was chosen as the study area.

2.In this manuscript, many sentences lack of citation, such as L66-69, L74-79, L112-117, L310-311, …. This is a very serious problem for an academic article. Please carefully go through the entire paper sentence by sentence, and add citations to each sentence that contains any knowledge from previous studies.

3.Please explain why you choose i-Tree in this study. And I’m curious if you used different model, will you get the same results? At least, give discussion of the uncertainties comes from model selection.

4.Table 1: Please add references of each model.

5.L73: Change the font of ‘Tang’. Please ensure consistent font usage throughout the paper.

6.L70-73: Instead of listing the related studies here, I think it would be better to summarize their key findings, advantages and disadvantages.

7.Figure 1 and Figure 2b: It is hard to distinguish the colors of ‘Park green space’ and ‘Square green space’ in the figure. Please consider changing the colors for clarity.

8.Figure 5: Please add names or description of x axis

9.L306-307: Why compare Kaifeng with Wuhan and Harbin instead of other cities? Please give some detailed explanation.

10.L320-321: In this paragraph, your statements of related studies and your work are about carbon storage, but you didn’t mention the results or key findings in ecology and climate. So, it would be better to briefly explain how the carbon storage contribute on regional ecology and climate.

Round 2

Reviewer 3 Report

Comments and Suggestions for Authors

The manuscript has been well revised and responded to  comments. I agree to accept the manuscript.

Comments on the Quality of English Language

The manuscript still needs some minor editing of the English language.

Reviewer 4 Report

Comments and Suggestions for Authors

Most of the comments are responded suitably. The manuscript has been improved significantly compared with the last version. I suggest accepting the current version.